# National Trends in Admissions, Treatments, and Outcomes for Dilated Cardiomyopathy (2016–2021)

**DOI:** 10.3390/medsci13030083

**Published:** 2025-06-23

**Authors:** Vivek Joseph Varughese, Abdifitah Mohamed, Vignesh Krishnan Nagesh, Adam Atoot

**Affiliations:** 1Department of Internal Medicine, University of South Carolina-Prisma Health, Columbia, SC 29201, USA; 2Department of Internal Medicine, University of Washington, Seattle, WA 98195, USA; drsharif@uw.edu; 3Department of Internal Medicine, Hackensack Palisades Medical Center, North Bergen, NJ 07047, USA; adamatoot.md@gmail.com

**Keywords:** dilated cardiomyopathy, heart transplantation, LVAD, CRT

## Abstract

Background: Dilated Cardiomyopathy (DCM) is one of the leading causes of non-ischemic cardiomyopathy in the United States (US). The aim of our study is to analyze the general trends in DCM admissions between 2016 and 2021, and analyze social and healthcare disparities in terms of treatments and outcomes. Methods: National Inpatient Sample (NIS) data for the years 2016 to 2021 were used for the analysis. General population trends were analyzed. Normality of data distribution was tested using the Kolmogorov–Smirnov test and homogeneity was assessed using Levine’s test. One-way ANOVA was used after confirmation of normality of distribution to analyze social and healthcare disparities. Subgroup analysis was conducted, with the paired *t*-test for continuous variables and Fischer’s exact *t*-test for categorical variables to analyze statistical differences. Multivariate regression analysis was conducted to analyze the association of factors that were significant in the one-way ANOVA and paired t/chi square tests. A two-tailed *p*-value < 0.05 was used to determine statistical significance. Results: A total of 5262 admissions for DCM were observed between 2016 and 2021. A general declining trend was observed in the total number of DCM admissions, with a 33.51% decrease in total admissions in 2021 compared to 2016. All-cause in-hospital mortality remained stable across the years (between 3.5% and 4.5%). A total of 15.3% of admissions had CRT/ICD devices in place. A total of 425 patients (8.07%) for DCM underwent HT, and 214 admissions for DCM (4.06%) underwent LVAD placements between 2016 and 2021 In terms of interventions for DCM, namely Cardiac Resynchronization Therapy (CRT), Left Ventricular Assist Devices (LVADs) and Heart Transplantations (HTs), significant variance was observed in the mean age of the admissions with admissions over the mean age of 55 had lower number of interventions. Significant variance in terms of sex was observed for DCM admissions receiving HT, with lower rates observed for females. In terms of quarterly income, patients belonging to the lowest fourth quartile had higher rates of LVAD and HT compared to general DCM admissions. In the multivariate regression analysis, age at admission had significant association with lower chances of receiving LVADs and HT among DCM admissions, and significant association with higher chances of all-cause mortality during the hospital stay. Conclusions: A general declining trend in the total number of DCM admissions was observed between 2016 and 2021. Significant gender disparities were seen with lower rates of females with DCM receiving LVADs and HT. DCM admissions with mean age of 55 and above were found to have significantly lower rates of receiving LVADs and HT, and higher chances of all-cause mortality during the admission.

## 1. Introduction

Dilated Cardiomyopathy (DCM) is a myocardial condition involving abnormal ventricular dilatation and systolic dysfunction in the absence of coronary artery diseases, valvular dysfunctions, long-standing arrhythmias, and conditions of increased afterload like uncontrolled hypertension. Mutations involving genes that code for cytoskeletal (intracellular microtubular structures) and nuclear envelope proteins account for about 35% of cases. Myocarditis, exposure to toxins (alcohol, drugs), and metabolic and endocrine disturbances are some of the acquired causes of DCM. Congestive Heart Failure (CHF) is the most common presenting symptom, although arrhythmias, circulatory collapse, as well as thromboembolic events have also been reported as the first presenting symptom of the condition [1]. Traditionally, cardiomyopathies were divided into ischemic and non-ischemic, and this definition holds utility in practical approach. However, with the advances made in molecular genetics, cardiomyopathies are now classified based on the primary organ involved [2]. Primary cardiomyopathies were defined as those solely or predominantly confined to heart muscle. Secondary cardiomyopathies had myocardial involvement as part of systemic diseases involving multiple organs like amyloidosis, sarcoidosis, hemochromatosis, as well as systemic toxicity arising from chemotherapy [3]. The European Society of Cardiology (ESC) Working Group on Myocardial and Pericardial Diseases classified heart muscle diseases based on their morphological and functional forms, and classified cardiomyopathies with further sub-stratification to familial and non-familial forms [4]. Most recently, the MOGE (S) nosology system was developed, which incorporates the morpho-functional phenotype (M) [considering both the structural and functional aspects of the myocardium], organ(s) involvement (O), genetic inheritance pattern (G), etiologic annotation (E) including genetic defect or underlying disease/substrate, and the functional status (S) of the disease using both the ACC/AHA HF stages and New York Heart Association (NYHA) functional class [5,6]. The natural history of DCM is hard to establish because of the heterogeneity in its presentation as well as differing rates in disease progression [7]. The onset of the disease can be insidious, especially in familial or idiopathic DCMs, and could be missed for a significant period of time [8]. The differential treatment benefit seen in DCM patients compared with patients with ischemic cardiomyopathy has been observed in several randomized clinical trials. However, these results were not consistently reproducible over subsequent trials, raising the question whether there is a true difference in response to treatment between ischemic and non-ischemic cardiomyopathies [9,10]. Currently, it is accepted that guideline-directed medical and device therapies, including implantable cardioverter-defibrillators (ICDs) and cardiac resynchronization therapy (CRT) for HF, are beneficial in DCM [1]. Heart Transplantation (HT) along with Mechanical Circulatory Support (MCS) systems are other advanced destination therapies in the management of DCM [11].

The aim of our study is to analyze the trends of admissions for DCM in the United States (US) between the years 2016 and 2021 using the National Inpatient Sample (NIS) for the corresponding years. Population demographics and treatment outcomes were also analyzed. Trends for ICD placements, LVADs, and HT among DCM admissions were analyzed for disparities in terms of age, gender, race and household income. Socioeconomic disparities in hospital outcomes for DCM admissions were analyzed.

## 2. Methods

The study has been designed as a retrospective cross-sectional longitudinal analysis. National Inpatient Sample data for the years 2016–2021 were used for the analysis. STATA was used for the statistical analysis. ICD 10 code I420 was used to select admissions with the admitting cause (I10_DX1) of Dilated Cardiomyopathy (DCM), and age more than 18. Stratification of population characteristics including sex, race, and age stratification was carried out and trends were analyzed over the years. Patients who died during the hospital stay were used to analyze the all-cause mortality over the years. Mean length of hospital stay and mean of total charges were analyzed over the years. Monthly quarterly income of the households was analyzed using the variable ZIPINC_QRTL of the NIS database (analyzed according to the respective ZIP codes). PCS 10 code 02YA0Z0 was used to select admissions that underwent Heart Transplantation (HT), and the proportion of admissions that underwent HT with admitting diagnosis of DCM was used to analyze the proportion of HT for DCM over the years. ICD 10 PCS code 02HA4QZ was used to select admissions that underwent Left Ventricular Assist Device (LVAD) placements over the years, and the proportion of admissions that underwent LVAD placements with admitting diagnosis as DCM were selected to analyze the proportion of LVAD placements in DCM.

Age, race, sex, and income stratification was carried out for the DCM admissions as well as LVAD/ HT for patients with an admitting diagnosis of DCM, to analyze social disparities in LVAD placements and HT for DCM. The Kolmogorov–Smirnov test was used to assess the normality of data distribution, and Levine’s test was used to analyze the homogeneity of variance, and one-way ANOVA was used to analyze the variance across the stratified groups to assess for significance in disparities. Mean difference was used for continuous variables and Pearson’s chi-squared test was used for categorical variables among patient factors used to assess for disparities.

Among factors that showed significant variance in the univariate analysis, multivariate regression models were used, accounting for age, sex, race, household income, and Charlson Comorbidity Index Score (a numerical scoring variable assigned to each admission based on comorbidities in the HCUP database) as well as APP DRG risk severity indices (numerical data assigned to admissions based on chances of in-hospital mortality in the HCUP database) [analyzed as independent variables, with the specific variable that showed significant variance used as the dependent variable]. A two-tailed *p*-value < 0.05 was used to determine statistical significance. Similar analyses of stratification were conducted for DCM admissions with CRT/ ICD devices as well as DCM admissions that died during the hospital stay. STATA 18.5 has been used for the statistical analysis.

## 3. Results

A total of 5262 admissions for Dilated Cardiomyopathy (DCM) were observed between 2016 and 2021. Population characteristics for admissions with admitting diagnosis of DCM are summarized in Table 1. National trends in total admissions for DCM are depicted in Figure 1. Population trends are depicted in Figure 2.

Analysis of DCM admissions with Cardiac Resynchronization Therapy (CRT) devices as well Implantable Cardioverter-Defibrillator (ICD) devices were analyzed, and results are depicted in Figure 3. Among the 5262 total admissions for DCM between 2016 and 2021, 803 admissions (15.3%) had CRT devices/ICD devices in place. Social disparities in the presence of ICD/CRT devices among DCM admissions are depicted in Table 2 and Figure 4, respectively.

Of the 5262 total admissions for DCM, 425 total heart transplantations were observed between the years 2016 and 2021.

Age, sex, race, and quarterly income stratification of general DCM admissions as well as DCM admissions that underwent heart transplantations, with analysis of disparities, are depicted in Table 3 and depicted in Figure 5.

Running the multivariate regression analysis, age at admission had a significant association with chances of receiving HT among DCM admissions, with an Odds Ratio of 0.962 (0.947–0.978), *p* value: 0.000, signifying increased age had significant association with lower chances of receiving HT.

Of the total DCM admissions, 214 admissions underwent LVAD placement between 2016 and 2021. Age, sex, race, and quarterly income stratification of general DCM admissions as well as DCM admissions that underwent LVAD placements, with analysis for population disparities, are depicted in Table 4 and depicted in Figure 6, respectively.

In the multivariate regression analysis, age at admission had significant association with chances of receiving LVAD placement among DCM admissions, with an Odds Ratio of 0.976 (0.959–0.994), *p* value: 0.010. Increased age was associated with significantly lower chances of receiving LVAD placements among DCM admissions.

Disparities between general DCM admissions and DCM admissions that died during the hospital stay were analyzed and are depicted in Table 5 and Figure 7, respectively.

In the multivariate regression analysis, age at admission had significant association with all-cause mortality during the admission, with an Odds Ratio of 1.030 (1.004–1.0565), *p* value: 0.021. Increased age was associated with significant chances of all-cause mortality among DCM admissions.

## 4. Discussion

Determining the incidence and prevalence of DCMs has been quite challenging because of geographic variations, patient selection, and changes in the diagnostic criteria. The prevalence of cardiomyopathy in underdeveloped and tropical countries is considerably higher than in developed countries. According to our analysis (Table 1 and Figure 1), a general declining trend in total admissions for DCM was observed. A 33.04% decrease in the total admissions for DCM was observed between 2016 and 2021. According to our analysis, the mean age of admissions for DCM remained between 53–55 years over the years, without any significant variance observed across the years. Male sex was observed to have a higher prevalence in the incidence of DCM, around 65%. Around 55% of the admissions for DCM were white, compared to 22–24% blacks, and no significant variance was observed over the years. All-cause in-hospital mortality of admissions remained at 3–4% without variance across the years. Around 30–35% of the admissions belonged to the lowest quartile of monthly income determined from respective ZIP codes.

Acute systolic congestive heart failure is the most common initial presentation for DCM. Guideline-Directed Medical Therapy (GDMT) for general CHF is the cornerstone in the symptomatic management of DCM. This includes Angiotensin-Converting Enzyme Inhibitors/Aldosterone Receptor Blocker Therapy (ACE/ARB), beta-blocker therapy, mineralocorticoid receptor antagonist therapy, digoxin, and ivabradine in refractory cases [12,13]. In patients with maximal tolerated GDMT for three months and ejection fraction remaining below 35%, implantable cardioverter-defibrillator therapy, and cardiac resynchronization therapy in the presence of left bundle branch block with QRS duration > 150 ms, is recommended [14,15]. According to our analysis, of the 5262 total admissions for DCM between 2016 and 2021, 803 admissions (15.3%) had CRT devices/ICD devices in place. This is depicted in Figure 3. The mean age of admissions for DCM that had ICD/CRT devices was higher (57.11 (56.09–58.14)) compared to the mean of general DCM admissions (54.67 (54.15–55.19)). Significant differences in patient characteristics were observed for age and race among DCM admissions that had ICD devices in place. A total of 14.149% of DCM admissions with documented white race had CRT devices in place, and 18.49% of DCM admissions with documented race as black had CRT devices in place during the admission. No patient factors had significant differences proven in the multivariate regression analysis for DCM admissions that had ICD/CRT devices in place. Results are summarized in Table 2 and Figure 4, respectively.

The American College of Cardiology, American Heart Association, and Heart Failure Society of America recommend Heart Transplantation (HT) for selected patients with advanced HF to improve survival and quality of life [16]. Reviewing the current literature, the logistical procedure for HT in DCM is not different from general heart transplantations. Table 3 and Figure 5 represent the trends in HT for DCM admissions between 2016 and 2021. Of the 5262 total admissions for DCM, 425 patients (8.07%) underwent HT between 2016 and 2021. The mean age for DCM admissions undergoing HT was significantly lower: 43.71 (41.45–45.97), compared to general DCM admissions: 54.67 (54.15–55.19). Significant variance was observed for age, sex, and quarterly income for DCM admissions undergoing HT, and the differences were proved with a paired *t*-test as well as Pearson’s chi-squared test. A total of 7.2% of the males with DCM diagnosis had documented HT during the follow-up period, while 5.67% was the prevalence for females. A total of 9.64% of the DCM admissions belonging to the highest quartile of monthly income had documented HT in the follow-up period, while only 4.47% of DCM admissions belonging to the lowest quartile of monthly income underwent HT in the follow-up period. Running the multivariate regression analysis, advanced age had significant lower association with chances of receiving HT among DCM admissions (OR: 0.962: 0.947–0.978, *p* value: 0.000). According to the International Society for Heart and Lung Transplantation (ISHLT) guidelines, there is a class I indication for patients under the age of 70 being considered for HT. As per the same guidelines, patients above 70 can be selected for HT when benefits outweigh risks [17]. In our analysis, the mean difference for ages of DCM admissions undergoing HT was 12.33 years lower than general DCM admissions. The mean age for DCM admissions undergoing HT was 43.71, and the comparing it to the mean age of DCM admissions that had documented all-cause mortality during the hospital stay, which is 57.3, a significant disparity is observed. This difference remained significant in the multivariate regression models after taking into account the Charlson comorbidity as well as the AP DRG risk severity indices. More real-life quality improvement studies need to be conducted on why a higher age becomes a prohibitive factor for HT in DCM patients, even when the mean age remains below 70. Socioeconomic factors need to be followed up in these evaluations, as DCM admissions belonging to the lowest quartile of monthly income had lower chances of undergoing HT when compared to highest income quartile.

A left ventricular assist device (LVAD) is employed in DCM as a destination therapy in refractory cases and for patients not meeting the criteria for HT [11]. The American College of Cardiology, American Heart Association, and Heart Failure Society of America recommend LVADs for patients with NYHA class IV symptoms who are dependent on intravenous inotropes or temporary mechanical circulatory support [18]. LVADs can also serve as a bridge to transplantation. Recent trials have shown a 2-year survival rate of over 80% with newer-generation LVADs, which approaches the early survival rates after cardiac transplantation [19,20]. A multidisciplinary team approach is recommended for evaluating and managing DCM patients undergoing LVAD implantation [21]. Trends for DCM admissions undergoing LVAD placements are shown in Table 4 and Figure 6, respectively. A total of 214 admissions for DCM (4.06%) underwent LVAD placements between 2016 and 2021. A similar trend in disparity analysis for HT was observed among LVAD placements for DCM, with significant differences observed in age and quarterly income. The mean age of DCM admissions undergoing LVAD placements was lower: 46.39 (43.65–49.13), compared to the mean age of general DCM admissions: 54.67 (54.15–55.19). Significant variance in terms of age was observed in the ANOVA and paired *t*-test. DCM admissions undergoing LVAD placements had a lower mean age by 9.89 years compared to general DCM admissions, and the significance stayed in multivariate regression models (OR: 0.967, 0.959–0.994, *p* value: 0.000). Similar to the trends seen for HT among DCM admissions, more quality investigations in real-life settings need to be conducted for increasing age being a prohibitive factor for LVAD placements. Like the trends with HT, a significant difference based on household income was observed among LVAD placements. A total of 6.17% of DCM admissions belonging to the highest quartile of monthly income had LVADs placed, while the percentage was 3.49% for DCM admissions belonging to the lowest quartile of monthly income. Economic disparities in HT and LVAD placements are well documented in the literature, and can be multifactorial, affecting not only access to transplantation, but also post-transplant outcomes. In a study by Kelty et al. [22], the barriers for low socioeconomic groups undergoing HT included reduced likelihood for being listed for transplantation and increased waitlist mortality. Increased waitlist mortality was associated with worsening clinical status and decreased post-transplant survival.

According to our analysis, the proportion of all-cause in-hospital mortality for general LVAD admissions remained stable over the years at 3–4%. The general mortality trends in DCM has shown a decreasing trend over the years, primarily due to the advanced heart failure therapies including MCS and HT in drug-refractory cases. The mean age of DCM admissions that died during the hospital stay was higher: 57.33 (54.57–60.08) compared to the mean age of general DCM admissions: 54.67 (54.15–55.19), with significant variance observed. Significant variance in terms of sex was observed in all-cause mortality among DCM admissions. A total of 74.48% of DCM admissions that died during the hospital stay were males, compared to 64.47% male admissions for DCM. A total of 35.53% of general DCM admissions were female, and the all-cause mortality among females was 25.12%. This is concurrent with a 2025 meta-analysis that proved significantly higher incidence and prevalence of DCM in males [23]. According to the American Heart Association, black individuals have almost a three-fold increased risk of developing DCM compared to white individuals. Black patients with DCM have a two-fold higher risk of mortality compared to age-matched white patients with DCM [2]. The genetics of DCM also differs between races. A study published in JAMA found that black patients with DCM are less likely to have clinically actionable genetic variants compared to white patients which can have a potential impact on clinical outcomes [24]. Healthcare and social disparities preventing access to advanced heart failure therapies are also postulated as a potential reason for worse outcomes in black patients. However, based on our analysis, no statistically significant variance was observed in terms of monthly income or race between general DCM admissions and DCM admissions that died during the hospital stay. 

## 5. Limitations of the Study

Although the National Inpatient Sample (NIS) is the largest publicly available database in the US with deidentified patient data, the database has its own limitations. ICD codes used at the time of discharge are used for the patient selection criteria, which is subject to inter-practitioner variability. This further limit patient selection for dilated cardiomyopathy. The database prevents us from following admissions over time; hence, longitudinal data could not be observed. In terms of prevalence of ICD/CRT use as well as LVAD placements and heart transplantations among DCM patients, stratification into patients meeting the criteria for each would have added more significance to the study. However, echocardiographic data for admissions are not coded in the NIS. Another potential limitation with the database is that some US hospitals do not participate in the NIS; hence, underestimation of the patient population is a factor to be kept in mind with the interpretation of the results. Another potential limitation is the lack of echocardiographic data to stratify patients meeting criteria for ICD/CRT devices.

## 6. Conclusions

Admissions for Dilated Cardiomyopathy (DCM) had a general declining trend across the years 2016 to 2021. A total of 5262 total admissions for DCM were observed across the years. A total of 15.3% of admissions had CRT/ICD devices in place. A total of 425 patients (8.07%) for DCM underwent HT, and 214 admissions for DCM (4.06%) underwent LVAD placements between 2016 and 2021. Analyzing the disparities, advanced age was associated with higher mortality among DCM admissions and had a lower chance of receiving treatment modalities like ICD/CRT devices, HT, and LVADs. More real-life qualitative improvement studies need to be conducted to analyze why age becomes a prohibitive factor for these advanced therapies among DCM patients, even when the mean age is under 70 years of age. Gender disparities were seen for DCM admissions receiving HT, with the rate being significantly lower in females. Significant economic disparities were noted for HT and LVAD placements among DCM admissions, with lower rates seen in the lowest income quartile groups.

## Figures and Tables

**Figure 1 medsci-13-00083-f001:**
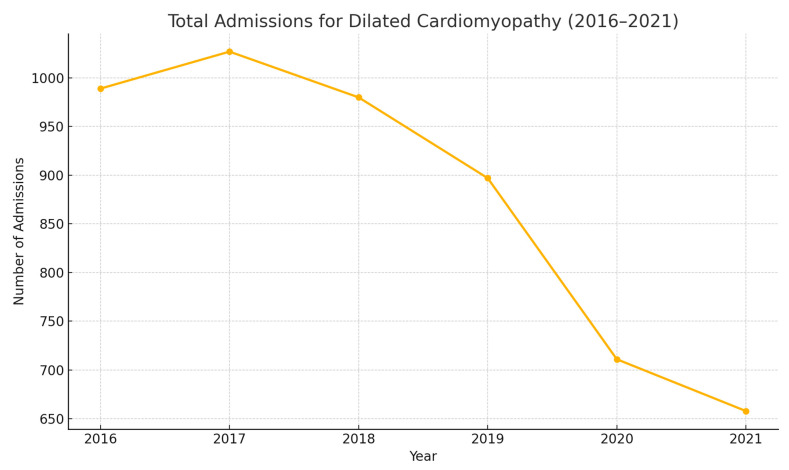
National trends in the total admissions for dilated cardiomyopathy (2016–2021).

**Figure 2 medsci-13-00083-f002:**
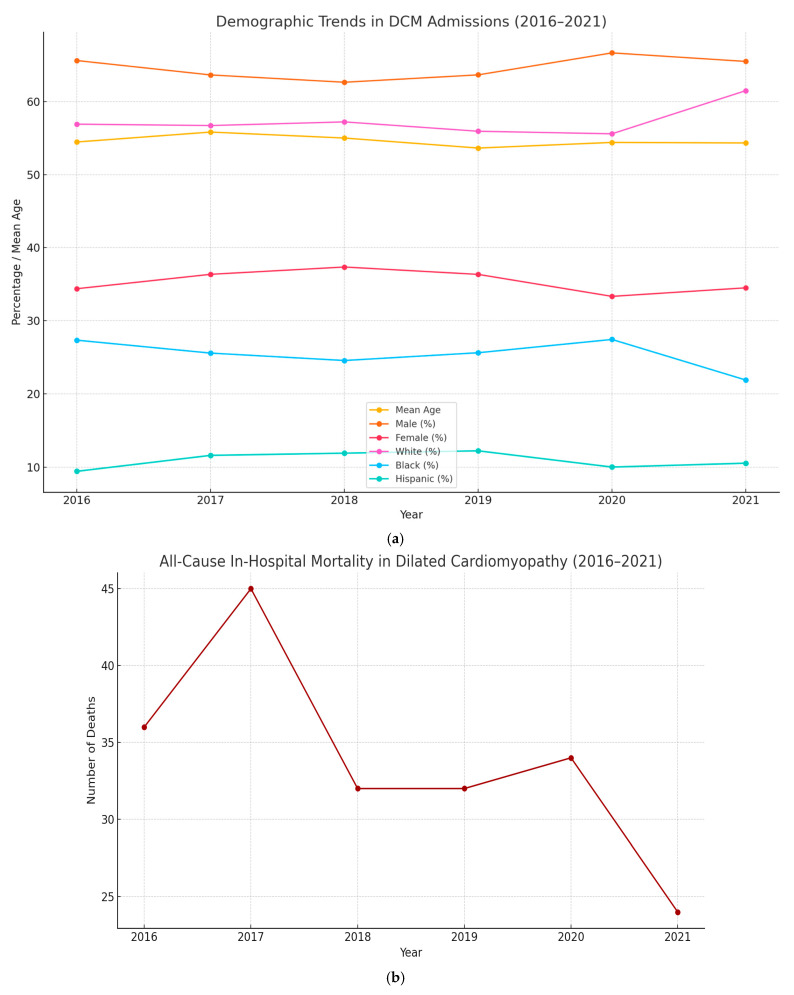
Population trends for dilated cardiomyopathy admissions (2016–2021). (**a**) Demographic trends for dilated cardiomyopathy admissions (2016–2021). (**b**) Trends in all-cause mortality for dilated cardiomyopathy admissions (2016–2021).

**Figure 3 medsci-13-00083-f003:**
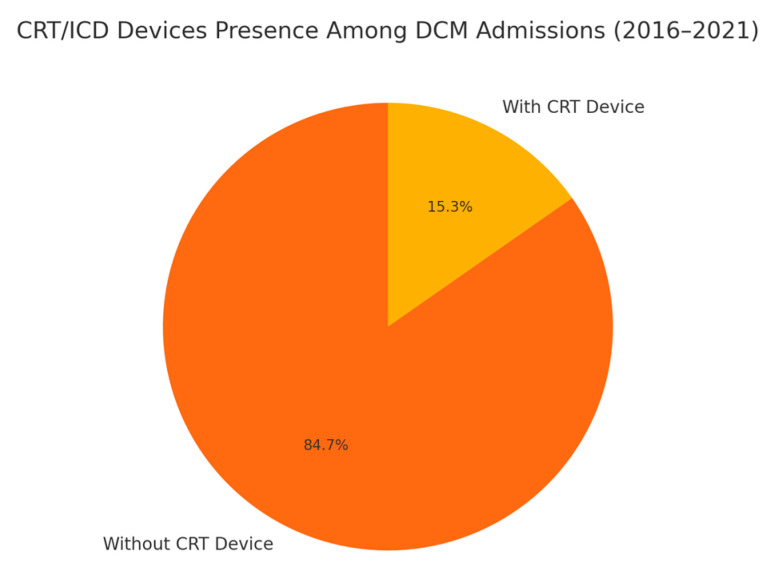
Dilated cardiomyopathy admissions with CRT/ICD devices.

**Figure 4 medsci-13-00083-f004:**
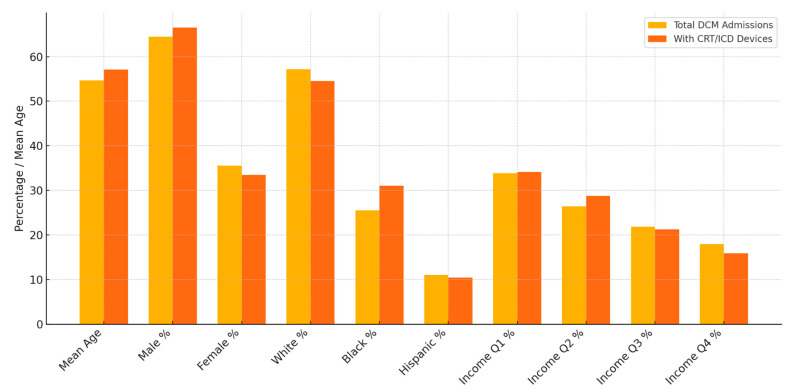
Analysis of disparities between general DCM admissions and DCM admissions with CRT/ICD devices.

**Figure 5 medsci-13-00083-f005:**
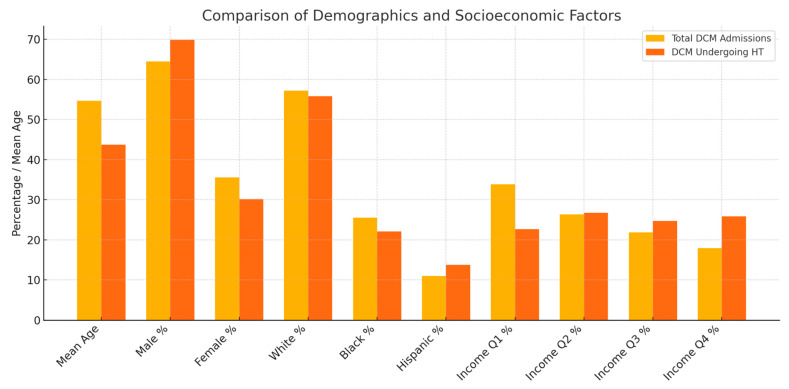
Analysis of disparities between general DCM admissions and DCM admissions undergoing HT.

**Figure 6 medsci-13-00083-f006:**
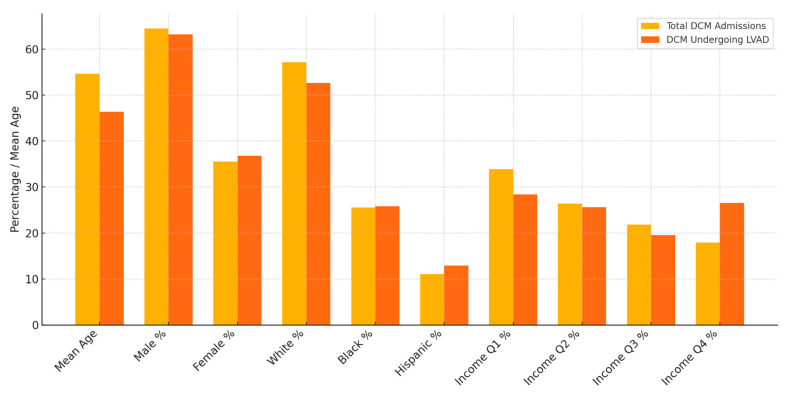
Analysis of disparities between general DCM admissions and DCM admissions undergoing LVAD placements.

**Figure 7 medsci-13-00083-f007:**
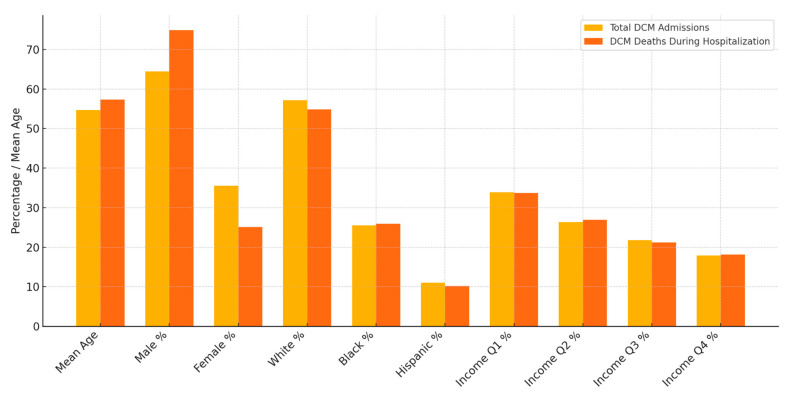
Analysis of disparities between general DCM admissions and DCM admissions who died during hospital stay.

**Table 1 medsci-13-00083-t001:** Population trends for dilated cardiomyopathy admissions (2016–2021).

Year	Total Number of Admissions for Dilated Cardiomyopathy	Mean Age	Sex	Race	All-Cause In-Hospital Mortality	Mean of Total Hospital Charges (USD)	Mean Length of Stay (Days)	Belonging to Lowest Quarter of Monthly Income (%)
2016	989	54.47 (53.28–55.66)	Male: 649 (65.62%) Female: 340 (34.38%)	White: 531 (56.91%)	36 (3.64%)	190,235.4	7.95 (6.83–9.06)	317 (32.92%)
				Black: 255 (27.33%)				
				Hispanic: 88 (9.43%)				
2017	1027	55.83 (54.67–56.98)	Male: 653 (63.65%) Female: 373 (36.35%)	White: 557 (56.72%)	45 (4.38%)	203,101.7	8.13 (7.09–9.17)	340 (34.10%)
				Black: 251 (25.56%)				
				Hispanic: 114 (11.61%)				
2018	980	55.01 (53.84– 56.19)	Male: 614 (62.65%) Female: 366 (37.35%)	White: 543 (57.22%)	32 (3.27%)	212,764.4	8.09 (7.01–9.16)	327 (34.28%)
				Black: 233 (24.55%)				
				Hispanic: 113 (11.91%)				
2019	897	53.64 (52.37–54.91)	Male: 571 (63.66%) Female: 326 (36.34%)	White: 485 (55.94%)	32 (3.57%)	213,565.3	8.15 (7.23–9.08)	331 (37.44%)
				Black: 222 (25.61%)				
				Hispanic: 106 (12.23%)				
2020	711	54.41 (52.94–55.89)	Male: 474 (66.67%) Female: 237 (33.33%)	White: 383 (55.59%)	34 (4.78%)	241,843.1	9.37 (7.80–10.93)	236 (33.67%)
				Black: 189 (27.43%)				
				Hispanic: 69 (10.01%)				
2021	658	54.34 (52.76–55.91)	Male: 431 (65.50%) Female: 227 (34.50%)	White: 385 (61.50%)	24 (3.65%)	311,854. 5	9.77 (8.10–11.45)	192 (29.63%)
				Black: 137 (21.88%)				
				Hispanic: 66 (10.54%)				

**Table 2 medsci-13-00083-t002:** Analysis of variance for social disparities (DCM versus DCM admissions with ICD/CRT).

	Total DCM Admissions	DCM Admissions with CRT/ICD Devices	Comparison of Differences
Mean Age	54.67 (54.15–55.19)	57.11 (56.09–58.14)	Two-sample *t*-test: Mean difference: −2.866, *p* value: 0.0001
Sex	Male: 3392 (64.47%)	Male: 534 (66.50%)	Pearson’s chi-squared test: 1.698, *p* value: 0.192
	Female: 1869 (35.53%)	Female: 269 (33.50%)	
Race	White: 2884 (57.15%)	White: 418 (54.50%)	Pearson’s chi-squared test: 20.46, *p* value: 0.001
	Black: 1287 (25.51%)	Black: 238 (31.03%)	
	Hispanic: 556 (11.02%)	Hispanic: 80 (10.43%)	
Quarterly Income (Quartile)	1st: 1743 (33.86%)	1st: 266 (34.15%)	Pearson’s chi-squared test: 4.225, *p* value: 0.238
	2nd:1358 (26.38%)	2nd: 224 (28.75%)	
	3rd: 1123 (21.82%)	3rd: 165 (21.28%)	
	4th: 923 (17.93%)	4th: 124 (15.92%)	

DCM: Dilated Cardiomyopathy; CRT: Cardiac Resynchronization Therapy; ICD: Implantable Cardioverter-Defibrillator.

**Table 3 medsci-13-00083-t003:** Analysis of variance for social disparities among general DCM admissions and DCM admissions undergoing HT.

	Total DCM Admissions	DCM Admissions Undergoing HT	Comparison of Differences
Mean Age	54.67 (54.15–55.19)	43.71 (41.45–45.97)	**Two-sample *t*-test:** **Mean difference: 12.33,** ***p* value < 0.0001**
Sex	Male: 3392 (64.47%)	Male: 245 (69.80%)	Pearson’s chi-squared test: 4.615, *p* value: 0.032
	Female:1869 (35.53%)	Female: 106 (30.20%)	
Race	White: 2884 (57.15%)	White: 182 (55.83%)	Pearson’s chi-squared test: 8.077, *p* value: 0.152
	Black: 1287 (25.51%)	Black: 72 (22.09%)	
	Hispanic: 556 (11.02%)	Hispanic: 45 (13.80%)	
Quarterly Income (Quartile)	1st: 1743 (33.86%)	1st: 78 (22.67%)	Pearson’s chi-squared test: 29.15, *p* value: 0.00
	2nd: 1358 (26.38%)	2nd: 92 (26.74%)	
	3rd: 1123 (21.82%)	3rd: 85 (24.71%)	
	4th: 923 (17.93%)	4th: 89 (25.87%)	

DCM: Dilated Cardiomyopathy, HT: Heart Transplantation.

**Table 4 medsci-13-00083-t004:** Analysis of variance for social disparities (general DCM versus LVAD placements in DCM).

	Total DCM Admissions	DCM Admissions Undergoing LVAD Placements	Comparison of Differences
Mean Age	54.67 (54.15–55.19)	46.39 (43.65–49.13)	**Two-sample *t*-test:** **Mean difference: 9.89,** ***p* value: 0.000**
Sex	Male: 3392 (64.47%)	Male: 139 (63.18%)	Pearson’s chi-squared test: 0.456, *p* value: 0.499
	Female: 1869 (35.53%)	Female: 81 (36.82%)	
Race	White: 2884 (57.15%)	White: 110 (52.63%)	Pearson’s chi-squared test: 7.573, *p* value: 0.181
	Black: 1287 (25.51%)	Black: 54 (25.84%)	
	Hispanic: 556 (11.02%)	Hispanic: 27 (12.92%)	
Quarterly Income (Quartile)	1st: 1743 (33.86%)	1st: 61 (28.37%)	Pearson’s chi-squared test: 8.0416, *p* value: 0.045
	2nd: 1358 (26.38%)	2nd: 55 (25.58%)	
	3rd: 1123 (21.82%)	3rd: 42 (19.53%)	
	4th: 923 (17.93%)	4th: 57 (26.51%)	

DCM: Dilated Cardiomyopathy, LVAD: Left Ventricular Assist Device.

**Table 5 medsci-13-00083-t005:** Analysis of variance for all-cause in-hospital mortality for DCM admissions.

	Total DCM Admissions	DCM Admissions That Died During the Hospital Stay	Comparison of Differences
Mean Age	54.67 (54.15–55.19)	57.33 (54.57–60.08)	**Two-sample *t*-test:** **Mean difference: 2.75,** ***p* value: 0.0465**
Sex	Male: 3392 (64.47%)	Male: 152 (74.88%)	Pearson’s chi-squared test: 9.9875 *p* value: 0.002
	Female: 1869 (35.53%)	Female: 51 (25.12%)	
Race	White: 2884 (57.15%)	White: 108 (54.82%)	Pearson’s chi-squared test: 8.509 *p* value: 0.130
	Black: 1287 (25.51%)	Black: 51 (25.89%)	
	Hispanic: 556 (11.02%)	Hispanic: 20 (10.15%)	
Quarterly Income (Quartile)	1st: 1743 (33.86%)	1st: 65 (33.68%)	Pearson’s chi-squared test: 0.0616 *p* value: 0.996
	2nd: 1358 (26.38%)	2nd: 52 (26.94%)	
	3rd: 1123 (21.82%)	3rd: 41 (21.24%)	
	4th: 923 (17.93%)	4th: 35 (18.13%)	

DCM: Dilated Cardiomyopathy.

## Data Availability

Data obtained using the HCUP National Inpatient Sample Database (publicly available deidentified database).

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
