# Peer review of "National Trends in Admissions, Treatments, and Outcomes for Dilated Cardiomyopathy (2016–2021)"

_medsci, 2025, doi:10.3390/medsci13030083_

Round 1

Reviewer 1 Report

Comments and Suggestions for Authors

The text includes a significant amount of medical jargon and complex terminology (e.g., "morphofunctional phenotype," "cytoskeletal and nuclear envelope proteins").

Simplifying this language or providing brief explanations for complex terms would make the content more accessible to a broader audience, including healthcare professionals who may not specialize in cardiology.

What factors contributed to the declining trend in DCM admissions from 2016 to 2021?

How do the outcomes for DCM patients who received Heart Transplantation or LVAD compare to those who did not?

What interventions could address the observed disparities in treatment based on household income for DCM admissions?

Author Response

The text includes a significant amount of medical jargon and complex terminology (e.g., "morphofunctional phenotype," "cytoskeletal and nuclear envelope proteins").

Reply: explanations added 

What factors contributed to the declining trend in DCM admissions from 2016 to 2021?

Reply: we were not able to find a suitable explanation for the trend based on our literature review. One thing to consider ( that we have added to the limitations section), is that some of the hospitals in the US does not pariticipate in the NIS, so underestimation could definitely be a factor

How do the outcomes for DCM patients who received Heart Transplantation or LVAD compare to those who did not?

Reply: a very good point. Unfortunately, the nature of the database only allows us to see what happened to an individual patient in the given admission and doesnt allow us to follow patients longitudinally 

What interventions could address the observed disparities in treatment based on household income for DCM admissions?

Reply: changes made 

Reviewer 2 Report

Comments and Suggestions for Authors

The authors present a comprehensive analysis on the trends in admissions, treatment, and outcomes in patients with dilated cardiomyopathy. The analysis included data between 2016 and 2021. The authors reveal that admissions for DCM had a general declining trend across the years. 5262 total admissions for DCM were observed across the years. Males were observed to have significantly higher numbers of DCM admissions as well as mortality during the hospital stay.  

The manuscript reveal interesting and comprehensive analysis. However, there are some issues deserving adressing.

  1. The structure of the abstract seems inappropriate. The conclusion section of the abstract includes results and is longer than all the other parts of the abstract.
  2. Could the authors add another figure illustrating the trend of mortality over the years?

Author Response

  1. The structure of the abstract seems inappropriate. The conclusion section of the abstract includes results and is longer than all the other parts of the abstract. Reply: abstract has  redone. Results section made more comprehensive 
  2. Could the authors add another figure illustrating the trend of mortality over the years?  Reply: added 

Reviewer 3 Report

Comments and Suggestions for Authors

Thank you for the opportunity to review the article titled “National Trends in Admissions, Treatments, and Outcomes for Dilated Cardiomyopathy (2016–2021)”.
The paper contains valuable data that deserves to be presented. However, I have a few questions and suggestions regarding certain aspects that, in my opinion, require improvement:

1. You mentioned applying Levene’s test for homogeneity of variances. Which test did you use to assess the normality of the data distribution?
2. The presentation of the tables is difficult to follow and would benefit from clearer formatting.
3. In Table 2, the way the data is presented suggests that you compared:
         Total DCM admissions – male vs. female and DCM admissions with CRT/ICD devices – male vs. female However, it is not clear whether these comparisons were actually performed.

The same issue applies to Tables 4 and 6.
4. Why did you choose ANOVA as the statistical analysis method? Do you consider it the most appropriate in this context?
5. It would be advisable to revise the statistical section within the Results chapter for improved clarity and methodological rigor.
6. The conclusions read more like a continuation of the results section. They should be rewritten to clearly highlight the implications and contributions of the study.

Author Response

  1. You mentioned applying Levene’s test for homogeneity of variances. Which test did you use to assess the normality of the data distribution? reply: kolmogorov-Smirnov test was used to assess the normality of distribution. This point has been added in the methods and abstract.
  2.  The presentation of the tables is difficult to follow and would benefit from clearer formatting. Reply: we have changed the tables with addition of chi square and t tests: is there any specific factors you would want to change ? 
  3. In Table 2, the way the data is presented suggests that you compared:
             Total DCM admissions – male vs. female and DCM admissions with CRT/ICD devices – male vs. female However, it is not clear whether these comparisons were actually performed. REPLY: yes, the percentages of male and female among total DCM admissions as well admissions undergoing ICD/LVAD/ HT: in the discussion section, percentage of each of the subsection ( ie male, whites, blacks, income strata) undergoing these treatments are separately mentioned, while discussing disparities. 
  4. Why did you choose ANOVA as the statistical analysis method? Do you consider it the most appropriate in this context?REPLY: we have added paires t test for continous variables ( age) and chi square for categorical variable comparison among groups: multivariate regression analysis has also been done on the data 
  5.  It would be advisable to revise the statistical section within the Results chapter for improved clarity and methodological rigor.: REPLY: changes made, kindly review 
  6.  The conclusions read more like a continuation of the results section. They should be rewritten to clearly highlight the implications and contributions of the study. REPLY: changes made, kindly review 

Reviewer 4 Report

Comments and Suggestions for Authors

In this study, the authors present the epidemiological aspects of a cohort of patients with dilated cardiomyopathy (DCM), revealing interesting data from a group of not common patients and are often excluded from large clinical trials.

In the abstract section, there is a significant disproportion between the results presented and the conclusions drawn. It should be limited to 250 words. Tables should not be cited.

Although the introduction discusses various aspects of DCM, the justification for this study is not specified. Why would it be important to analyze the admission trends of these patients? What is your hypothesis and what would be your contribution?

The methodology should mention the type of study and how the subjects were selected.

The software used in the statistical analysis should be mentioned in the final paragraph of this section.

This section should have a subsection on definitions, which is essential in retrospective analyses. Beyond mentioning the codes, each of the variables analyzed in the patients must be defined or described.

You should simplify the presentation of the results. It is not necessary to present a table and figure for each finding; this makes the description redundant. One option would be a table summarizing all the outcomes or a figure with multiple panels.

Why is ANOVA used in comparisons of values? Most of the variables compared are qualitative and two groups’ comparisons. I have doubts about the suitability of this test for this purpose.

The discussion section is very deficient; it merely repeats the findings of the results. It does not mention the contributions of its study or comparisons with other series, nor does it even explain a reason or hypothesis for the decrease in admission rates. These aspects should be explained more clearly.

The limitations of the study should come before the conclusion.

Probably simplifying the results presented will allow you to write a simpler and more concrete discussion.

Author Response

In the abstract section, there is a significant disproportion between the results presented and the conclusions drawn. It should be limited to 250 words. Tables should not be cited.

REPLY: abstract has been reformatted. Kindly review 

Although the introduction discusses various aspects of DCM, the justification for this study is not specified. Why would it be important to analyze the admission trends of these patients? What is your hypothesis and what would be your contribution?

REPLY: since it is a trend analysis, we did not have null hypothesis to begin with as we were not looking for a specific factor, rather than checking trends for social disparities, especially in terms of treatment options and outcomes for DCM. This has been added to the introduction section. 

The methodology should mention the type of study and how the subjects were selected.

The software used in the statistical analysis should be mentioned in the final paragraph of this section.

REPLY: the type of study ( longitudinal retrosepctive cross sectional analysis and inclusion criteria mentioned. So is the software

This section should have a subsection on definitions, which is essential in retrospective analyses. Beyond mentioning the codes, each of the variables analyzed in the patients must be defined or described.

REPLY: definitions of the ZIPINC_QRTL ( quarterly income), APDRG risk severity indices and charleson comorbdity indices have been explained 

You should simplify the presentation of the results. It is not necessary to present a table and figure for each finding; this makes the description redundant. One option would be a table summarizing all the outcomes or a figure with multiple panels.

REPLY: he was removed 2 tables and 2 figures that was not specifically adding much to the study. Kindly review 

Why is ANOVA used in comparisons of values? Most of the variables compared are qualitative and two groups’ comparisons. I have doubts about the suitability of this test for this purpose.

REPLY: entire stats section has been redone: paired t test as well chi square along with regression models have been included. Kindly review 

The discussion section is very deficient; it merely repeats the findings of the results. It does not mention the contributions of its study or comparisons with other series, nor does it even explain a reason or hypothesis for the decrease in admission rates. These aspects should be explained more clearly.

Reply: while we were not able to find a suitable explanation for the declining trends, we did reframe the discussion section, stratifying the resyults, implications and further prosepects: kindly review: extra references have been added 

The limitations of the study should come before the conclusion.

REPLY: changes made 

Probably simplifying the results presented will allow you to write a simpler and more concrete discussion.

REPLY: kindly review and let us know if any specific changes other than the ones made need to be done 

Round 2

Reviewer 2 Report

Comments and Suggestions for Authors

The authors revised the manuscript. I highly recommend that when you prepare the answers to the reviewers, provide a comprehensive full sentence reply. Furthermore, when adding a new figure or table, you should note the number of the figure and where in the revised text it can be found. Otherwise, the changes that have been made in the revised manuscript cannot be appreciated by the reviewers.

Author Response

The authors revised the manuscript. I highly recommend that when you prepare the answers to the reviewers, provide a comprehensive full sentence reply. Furthermore, when adding a new figure or table, you should note the number of the figure and where in the revised text it can be found. Otherwise, the changes that have been made in the revised manuscript cannot be appreciated by the reviewers.

Reply: sorry for the mistake from our side. In the earlier review, two main things were mentioned: 

  1. Structure of the abstract: we revised the abstract entirely (the new abstract in the resubmitted version is completely changed from the initial submission, hence we did not highlight it as a whole. Kindly review the modified abstract. We are highlighting the changed aspect of the abstract in this version (the conclusion has been toned down and more details have been added to the results section of the abstract as suggested by the reviewer)
  2. A figure representing the overall mortality was asked to be included: it is included as Figure 2b

Reviewer 4 Report

Comments and Suggestions for Authors

The authors have made considerable improvements to the structure and writing of the manuscript.

The suggestions were taken into account in the Abstract – Introduction section.

In terms of methodology, I believe the study design is more suited to a retrospective longitudinal analysis.

Remove the STATA specification on line 101.

Separate the Methodology section into paragraphs for easier reading.

Results: I find no differences between the titles of Figure 1 and Figure 2. Are Figure 2 the national data for patients with this diagnosis? Please explain further in the text.

Figure 1 has two figures. Please explain these results further.

Table 2 is missing the column DCM admissions WITHOUT CRT/ICD Devices. Comparisons should be between the groups WITH and WITHOUT CRT/ICD Devices,...

I don't know why the ANOVA test value is maintained.

Table 2 shows the same data as Figure 4. I don't see the point in repeating the data.

Table 3 shows the same data as Figure 5. Repeating data is unnecessary.

Specify in the methodology section how the regression models were run, as only the results are reported, not how they were created. Dependent variable? Independent variables?

The use of the comorbidity index is not specified in either the methodology or the results.

The discussion is unnecessarily long, repeating the findings of the results. It should focus on comparing your results and explaining the reasons for your results.

Author Response

In terms of methodology, I believe the study design is more suited to a retrospective longitudinal analysis.

Reply: this has been mentioned in the methods section. Since there are considerable changes in formatting happening during conversion of the word files between systems, we are highlighting the changed aspects in red color for easiness of review. 

Remove the STATA specification on line 101.

Reply: the specification has been removed 

Separate the Methodology section into paragraphs for easier reading.

Reply: methods section has been reformatted 

Results: I find no differences between the titles of Figure 1 and Figure 2. Are Figure 2 the national data for patients with this diagnosis? Please explain further in the text.

reply: figure 1 is a graphical representation of " total admissions for DCM" between the years: this was more to serve as to see the general trend in total admissions every year 

While Figure 2a is the population trend for these admissions: ie the trends in how the admissions changed based on age, sex, race over the years. Figure 2 b are the trends in mortality among DCM admissions over the years ( this has been specified in the description of the images) 

Figure 1 has two figures. Please explain these results further.

Reply: it has split into 2a and 2 b and descriptions have been added 

Table 2 is missing the column DCM admissions WITHOUT CRT/ICD Devices. Comparisons should be between the groups WITH and WITHOUT CRT/ICD Devices,...

Reply: We agree with the comment. However, we used a statistical methods approach to see how the subpopulation among DCM admissions getting appropriate therapy differed from the general admissions, and to look for disparity in terms of getting appropriate treatment 

I don't know why the ANOVA test value is maintained.

Reply:  we used the factors having significant variance in the multivariate regression models. But if it's making the data look more cumbersome, we would just mention it in the methods section and omit it from the tables. 

Table 2 shows the same data as Figure 4. I don't see the point in repeating the data.

Table 3 shows the same data as Figure 5. Repeating data is unnecessary.

Reply: adding graphical representation to textual data was the idea behind this, as some reader prefer pictorial representation over busy tables. The other reviewer for the study was in fact mentioning adding more figures, updates existing ones: hence would leave up to the chief editor since there has been a conflict in suggestions. From our side, whatever best suites the journal layout and trends would be fine, as long as effective communication of results with readers are assured. 

Specify in the methodology section how the regression models were run, as only the results are reported, not how they were created. Dependent variable? Independent variables?

Reply: it has been clearly mentioned in the methods section how the regression models were used. We are highlighting it in red for easier review. The ANOVA was used to determine population factors having significant variance among the receival of treatment modalities ( CRT, transplant, LVADs etc). Age, sex, race, income, and comorbidity scores were used as independent variables in the analysis ( this was not mentioned earlier, as we assumed it would be understood, but adding text within parenthesis specifying the same). 

The use of the comorbidity index is not specified in either the methodology or the results.

Reply: the two comorbidity indices used: charleson score as well as AD DRG risk indices was  explained in the methods section ( on how it is coded within the database): is highlighted and made in bolds now

The discussion is unnecessarily long, repeating the findings of the results. It should focus on comparing your results and explaining the reasons for your results.

Reply: the discussion section involved chart review in the trends of all three interventoiones, which we felt as necessary to add context for disparities when it comes to DCM admissions: if the reviewer can specify what aspects of the discussion in unwanted, would be easier for us to omit/ alter/concise it